# The adaptor protein TASL is required for age-related B cell emergence and lupus-like disease development in mice

Julia C. Johnstone[1], Robert Mitchell[1], Timothy J. Vyse[2], Alexander J. Clarke [ID][1]*

1 Kennedy Institute of Rheumatology, University of Oxford, Oxford, United Kingdom, 2 Department of Medical and Molecular Genetics, School of Basic and Medical Biosciences, King's College London, London, United Kingdom

* alexander.clarke@kennedy.ox.ac.uk

## Abstract

The autoimmune disease systemic lupus erythematosus (SLE) is associated with genetic variants in the X-linked gene *CXORF21*, which encodes the protein TASL. TASL acts as an adaptor in the IRF5 pathway and is necessary for the phosphorylation of IRF5 in response to TLR7 or TLR9 stimulation. Here, we investigate the role of TASL in the humoral immune response, and in the development of lupus in the B6.MRL[lpr] murine model of SLE. We find that while TASL is dispensable for their development, it is required for the full activation of B cells via TLR9 stimulation, and consequent interferon signaling and inflammatory cytokine expression. Additionally, TASL is crucial for the emergence of age-associated B cells (ABCs), a B cell population derived from the extrafollicular response that increases with age and is expanded in autoimmune disease, and the production of IgG2c antibodies. We also find that deletion of TASL prevents the onset of autoimmunity in the genetically-determined B6.MRL[lpr] model of lupus.

## Introduction

Systemic lupus erythematosus (SLE) is a chronic autoimmune disease characterized by auto-antibodies against nuclear antigens. It is highly sexually dimorphic, and around 9 times more prevalent in females than in males [1]. Genome wide association studies (GWAS) identified *CXORF21*, which encodes the protein TLR Adaptor Interacting With Endolysosomal SLC15A4 (TASL), as a risk locus for SLE, and there is evidence that this gene may be linked to sexual dimorphism in SLE [2,3]. *CXORF21* is an interferon-response gene, and is encoded on the X-chromosome. *CXORF21* can escape X-inactivation, and its expression correlates with disease activity in young females with SLE [2]. The risk variant in *CXORF21* associated with SLE has also been shown to lead to increased expression of TASL, and potentially heightened responses to endolysosomal TLR stimulation [4].

which permits unrestricted use, distribution, and reproduction in any medium, provided the original author and source are credited.

**Data availability statement:** RNA sequencing data has been deposited on GEO with the accession number: GSE302697. The data underlying this manuscript can be found at: DOI 10.5281/zenodo.18649676.

**Funding:** Funding for this work was provided by the Wellcome Trust (211072/Z/18/Z) and the Medical Research Foundation (MRF-159-0005-ELP-CLAR-C0953) to A.J.C. and the Kennedy Trust for Rheumatology Research to J.C.J. and R.M. The funders had no role in study design, data collection and analysis, decision to publish, or preparation of the manuscript.

**Competing interests:** The authors have declared that no competing interests exist.

**Abbreviations:** ABCs, age associated B cells; AID, activation-induced cytidine deaminase; ANAs, anti-nuclear antibodies; BLN, brachial lymph nodes; GC, germinal center; GO, gene ontogeny; GSEA, gene set enrichment analysis; GWAS, genome wide association study; ISG, interferon stimulated gene; IMQ, imiquimod; LPS, lipopolysaccharide; MZ, marginal zone; NP-CGG, 4-hydroxy-3-nitrophenylacetyl -chicken gamma globulin; PBS-T, PBS-Tween20; pDCs, plasmacytoid dendritic cells; SHM, somatic hypermutation; SLE, systemic lupus erythematosus; SRBC, sheep red blood cells; XCI, X-chromosome inactivation.

TASL acts as an adaptor between endolysosomal TLR signaling and the IRF5 pathway [5]. Endolysosomal TLR signaling via TLR7 and TLR9, which respond to RNA and CpG-rich DNA respectively, plays an important role in the development of SLE, and there are also many risk variants in *IRF5* associated with SLE [3]. TASL is located on the endolysosomal membrane and associates with SLC15A4, a histidine transporter that is also genetically associated with SLE [6–8]. B cell-specific deletion of *SLC15A4* impairs endolysosomal TLR signaling, and ameliorates clinical features of SLE and production of auto-antibodies in a murine model [9]. Whilst it was initially thought that the transporter role of SLC15A4 was critical, more recently it has been demonstrated that its adaptor function is of more importance [10].

The absence of TASL prevents IRF5 phosphorylation, and thereby downstream interferon stimulated gene (ISG) expression and proinflammatory cytokine production [11]. TASL has been shown to play a role in the activation and function of B cells, and deletion of *TASL* in chemically-induced models of lupus has been shown to decrease development of autoimmunity [4,11]. However, questions still remain about the role TASL plays in humoral responses, and in other more representative models of lupus.

B cells differentiate via two main pathways depending on the stimulus: the germinal center (GC) reaction and the extrafollicular response. In the GC reaction B cells undergo rapid proliferation and somatic hypermutation (SHM). In SHM, mutations in the immunoglobulin locus are introduced by activation-induced cytidine deaminase (AID), and then competition for T cell help to survive and proliferate occurs, leading to improvements in antigen binding, a process termed affinity maturation. In the extrafollicular response, T-independent stimulation results in the development of short-lived plasmablasts at extrafollicular sites [12].

In lupus, both the formation of spontaneous GCs and an aberrant extrafollicular response are seen, and the contribution of each of these to the disease is debated [13]. Increasing evidence suggests that the extrafollicular response plays a critical role, and one element of this is the expansion of age-associated B cells (ABCs), a subset predominantly produced extrafollicularly. ABC frequency is correlated with disease severity and autoantibody levels in SLE patients [14], and they play a role in the pathogenesis of lupus in mice [15]. IRF5 deficiency prevents the development of ABCs [16,17], but whether TASL is also required and is similarly important for the extrafollicular response is not known.

We show that TASL plays a critical role in the activation of B cells and the production of plasmablasts in response to immunization. Additionally, we find that the absence of TASL prevents SLE in a genetically-determined murine model, abrogating the development of auto-antibodies and production of ABCs.

## Results

### *Tasl* deletion prevents full activation of B cells via TLR9

To investigate the role of TASL in vivo, we generated *Tasl* [KO] mice (S1A Fig). Steady-state peripheral populations of B cell subsets, as well as T cells, macrophages, and plasmacytoid dendritic cells (pDCs) were unaffected by the absence of TASL (S1B–S1M Fig). B cell development in the bone marrow was also normal (S1N–S1O Fig).

We first stimulated B cells in vitro with either lipopolysaccharide (LPS), imiquimod (IMQ), or CpG, agonists for TLRs 4,7, and 9 respectively. Viability and proliferation were not substantially impacted by the absence of TASL following stimulation with any of the agonists (S2A Fig). However, we observed decreased levels of the surface activation marker CD69 upon IMQ or CpG stimulation, but not with LPS, in both male and female mice (Figs 1A and S2B). Interestingly however, expression of the co-stimulatory molecule CD86 was not reduced, and was in fact relatively increased in TASL-deficient B cells following CpG stimulation (Figs 1B and S2B).

A higher proportion of B cells remained IgD⁺ when stimulated by CpG (Figs 1C,1D, and S2D), reflecting persistence of a naïve state. TASL-deficient B cells expressed lower levels of the transcription factor *Prdm1*, which encodes Blimp1, required for plasmablast differentiation, following CpG stimulation (Fig 1E). In keeping with this finding, loss of TASL led to a lower proportion of CD138⁺IRF4⁺ plasmablasts following stimulation with CpG (Figs 1F,1G and S2E).

We next performed RNA sequencing on CpG-stimulated *Tasl* $^{KO}$ and wild type B cells. We identified 633 differentially expressed genes, and performed gene set enrichment analysis (GSEA) (Fig 1H, 1I). We found downregulation of key ISGs, such as *Irf7* and *Ifi44,* and the interferon-mediated signaling pathway in *Tasl* $^{KO}$ B cells. Using qPCR we confirmed reduced expression of *Irf7* and the ISGs *Isg15* and *Mx1*, and cytokines *Il6* and *Il10* (Fig 1J–1N). In keeping with our finding of elevated CD86 with CpG stimulation following *Tasl* deletion, the antigen processing and presentation pathway was upregulated.

TASL is therefore required for B cell activation and plasmablast differentiation in vitro following TLR9 stimulation.

## Loss of TASL impairs humoral immunity

To understand whether TASL is important in humoral immunity, we immunized *Tasl* $^{KO}$ and wild type mice with 4-hydroxy-3-nitrophenylacetyl-chicken gamma globulin (NP-CGG), a protein-hapten antigen that initiates a T-dependent immune response. At 14 days post-immunization we found a reduction in the proportion and number of splenic GC B cells in *Tasl* $^{KO}$ mice (Fig 2A, 2B). Plasmablasts were also less abundant, but notably only in female mice (Fig 2C, 2D). This was associated with a reduction in resting serum IgG2b levels in female *Tasl* $^{KO}$ mice, slightly lower levels of other immunoglobulin isotypes, and reduced free kappa and lambda light chains (S3A Fig). No differences in resting immunoglobulins were observed in male mice. In response to NP-CGG immunization, there was no difference in levels of anti-NP IgG1 antibodies binding to either high (NP₂₀) or low (NP₂) conjugation-ratio proteins, reflecting low and high affinity antibodies respectively (Fig 2E). Antibody affinity maturation, measured as the ratio between NP₂ and NP₂₀ binding, was also unaffected by loss of TASL (Fig 2E). However anti-NP IgG2c antibody levels were markedly lower in both male and female *Tasl* $^{KO}$ mice, with lower IgG2c production in the former group confirmed using ELISpot (Fig 2F and 2G).

The lack of difference in levels of anti-NP IgG1 antibodies, with no impairment of affinity maturation, suggested that TASL may be dispensable for the GC reaction. To test this, we generated competitive mixed bone marrow chimeras, using CD45.1 wild type and CD45.2 wild type or *Tasl* $^{KO}$ bone marrow cells. Following reconstitution, mice were immunized with sheep red blood cells (SRBC), a T-dependent antigen, and the GC response examined at day nine. The relative proportion of wild type and *Tasl* $^{KO}$ B cells in the GC reaction and in the plasmablast population was not different (Fig 2H, 2I). We also generated in vitro GC B cells (iGCBCs) using the iGB system, in which B cells are cultured on a fibroblast layer which expresses BAFF and CD40L, in the presence of IL-4, for 5 days [18]. We did not detect a difference in the capacity of wild type and *Tasl* $^{KO}$ B cells to differentiate into iGCBCs (S4A–S4B Fig).

We then immunized mice with NP-Ficoll, a T-independent antigen which elicits a predominantly extrafollicular response [19], characterized by IgM antibodies. We found that *Tasl* $^{KO}$ mice of both sexes had a markedly reduced level of anti-NP IgM antibodies at day seven following immunization (Fig 2J), and decreased plasmablast formation was seen in female mice (Fig 2K). Using a competitive bone marrow chimera, as before with CD45.1 wild type, and either wild type or *Tasl* $^{KO}$ CD45.2 bone marrow cells, following NP-Ficoll immunization there was a reduction in the proportion of *Tasl* $^{KO}$ plasmablasts compared to CD45.1 wild type competitors at day 7 (Fig 2L).

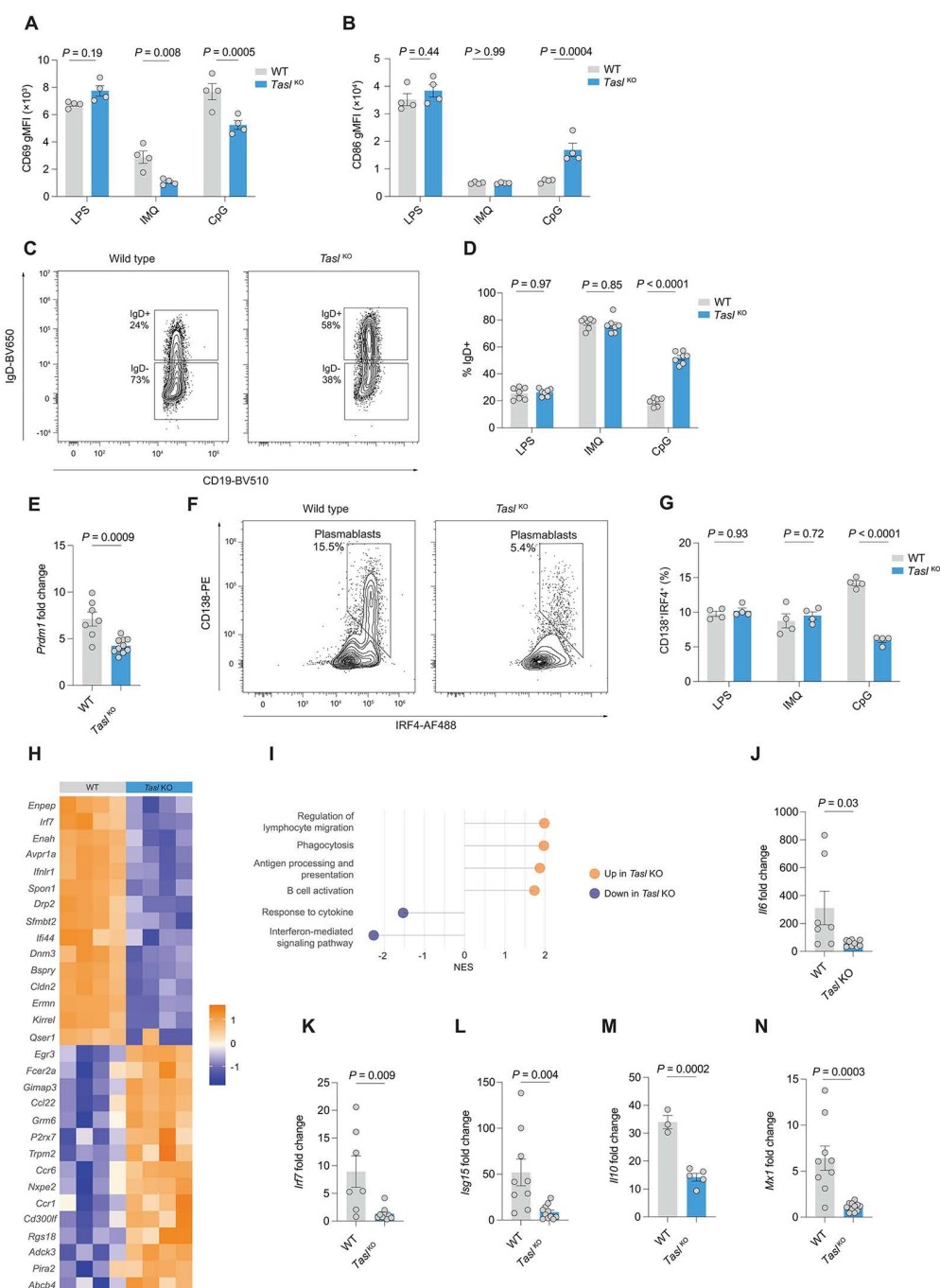

**Fig 1. Tasl deletion prevents full activation of B cells via TLR9. A**. Flow cytometric measurement of CD69 geometric mean fluorescence intensity (gMFI) in splenic B cells from either WT (*n* = 4) or Tasl KO (*n* = 4) female mice after stimulation with either lipopolysaccharide (LPS), imiquimod (IMQ), or CpG for 24 hours. Representative of two independent experiments. **B**. Flow cytometric measurement of CD86 gMFI in splenic B cells from either WT (*n* = 4) or *Tasl* KO (*n* = 4) female mice after stimulation with either LPS, IMQ, or CpG for 24 hours. Representative of two independent experiments. **C**. Representative flow cytometry gating of IgD+ and IgD− CD19+ B cells from WT and *Tasl* KO mice following stimulation with CpG for 72 hours. **D**. Quantification of IgD+ B cells as percentage of total B cells, after stimulation of isolated splenic B cells from female WT (*n* = 7) or *Tasl* KO (*n* = 7) mice with either LPS, IMQ, or CpG for 72 hours, as in C. Representative of and pooled from 2 independent experiments. **E**. Expression of *Prdm1* after stimulation of isolated splenic B cells from female WT (*n* = 7) or *Tasl* KO (*n* = 10) mice with CpG for 24 hours. Representative of and pooled from 3 independent experiments. **F**. Representative flow cytometric gating of CD19+IgD−CD138+IRF4+ plasmablasts differentiated from WT or *Tasl* KO B cells after 72 hours simulation with

CpG. **G**. Quantification of plasmablasts as CD19+IgD−CD138+IRF4− B cells (as in F) after culture of B cells with either LPS, IMQ, or CpG for 72 hours (*n* = 4). Representative of 2 independent experiments. **H**. Heatmap of top differentially expressed genes by RNA sequencing of WT or *Tasl* KO B cells stimulated with CpG for 24 hours (*n* = 4). **I**. Gene set enrichment analysis (GSEA) of RNA sequencing as in H. All pathways shown are statistically significant (adjusted *P*-value < 0.05). Normalized enrichment score for gene ontogeny (GO) terms shown. **J–N** Expression of *Il6, Irf7, Isg15, Il10*, and *Mx1* measured by qPCR after stimulation of B cells from female WT (*n* = 7) or *Tasl* KO (*n* = 10) mice with CpG for 24 hours. Representative of and pooled from 2 independent experiments. Statistical significance was calculated using two-way ANOVA with Šidák's multiple testing correction (A, B, D, G) or two-tailed unpaired *t* test (E, J–N). Data are presented as mean ± S.E.M. Each point represents a single mouse. The data underlying this figure can be found at: DOI https://doi.org/10.5281/zenodo.18649676.

Together these data demonstrate that TASL is primarily required for the extrafollicular response, with a limited role in the GC reaction.

### TASL is required for the formation of age-associated B cells

Age-associated B cells (ABCs) are a B cell subset which accumulates with age and in autoimmune disease, and are proportionately much more frequent in female mice and humans [17,20,21]. ABCs are of predominantly extrafollicular origin, and we therefore examined whether TASL was required for their accumulation. We found that as expected, ABCs were more abundant in aged female wild type mice, but were markedly reduced in *Tasl* KO mice of both sexes at eight months of age (Fig 3A–3C).

To determine whether this effect was cell intrinsic, we cultured isolated B cells from wild type and *Tasl* KO mice with the TLR7 agonist IMQ, agonistic anti-CD40, anti-IgM F(ab')$_2$, and IL-21 to induce ABC differentiation (Fig 3D) [22–24]. ABC formation was substantially reduced in B cells from *Tasl* KO mice. ABCs are potent producers of IgG2c antibodies, and its level in the culture supernatant was also much decreased (Fig 3E–G).

TASL is therefore required for the emergence of ABCs in aged mice, and following culture in vitro with ABC-inducing stimuli.

### Absence of TASL prevents the development of lupus in B6.MRL<sup>lpr</sup> mice

We next examined whether TASL was necessary for the development of lupus. We crossed *Tasl* KO mice with the B6.MRL-*Fas*<sup>lpr</sup>/J congenic lupus mouse model, generating B6.MRL<sup>Lpr</sup> × *Tasl* KO mice (hereafter referred to as MRL<sup>Lpr</sup> × *Tasl* KO). These mice were aged to 20 weeks to allow disease to develop, and compared with B6.MRL<sup>lpr</sup> mice, or wild type C57BL/6 controls.

We found that the absence of TASL decreased splenomegaly and lymphadenopathy to wild type levels (Fig 4A–4C). The spontaneous GCs seen in MRL<sup>Lpr</sup> mice were absent in MRL<sup>Lpr</sup> × *Tasl* KO mice (Fig 4D–4F), and the frequency and count of plasmablasts was also decreased, to a larger extent in females than in males (Fig 4G–4H). There was also a marked reduction in effector (CD62L<sup>lo</sup>CD44<sup>high</sup>) T cells and increased numbers of naive T cells (CD62L<sup>hi</sup>CD44<sup>low</sup>) (Fig 4I, 4J), but this was only seen in female mice, although a trend in the same direction was observed in males. We also observed no difference in the frequencies or numbers of pDCs in MRL<sup>Lpr</sup> × *Tasl* KO compared to MRL<sup>Lpr</sup> mice, but they were less activated, with a smaller proportion CD86<sup>+</sup> (S5A–S5C Fig).

ABCs are a key element of the pathogenesis of lupus [14,15,25]. We therefore investigated whether the absence of TASL impacted the development of this population in MRL<sup>Lpr</sup> × *Tasl* KO mice.

We found that CD11c<sup>+</sup>T-bet<sup>+</sup> ABCs were almost completely absent in the spleens of both male and female MRL<sup>Lpr</sup> × *Tasl* KO mice (Fig 4K–4M). In addition, MRL<sup>Lpr</sup> × *Tasl* KO mice had drastically reduced anti-nuclear antibodies (S5D Fig), and much lower levels of IgG2c anti-dsDNA antibodies (S5E Fig). Anti-Sm/RNP antibodies were also decreased, but only significantly in female mice (S5F Fig).

Deletion of *Tasl* therefore abrogates the development of clinical and immunologic features of lupus and is required for the formation of ABCs in the B6.MRL-*Fas*<sup>lpr</sup>/J mouse model.

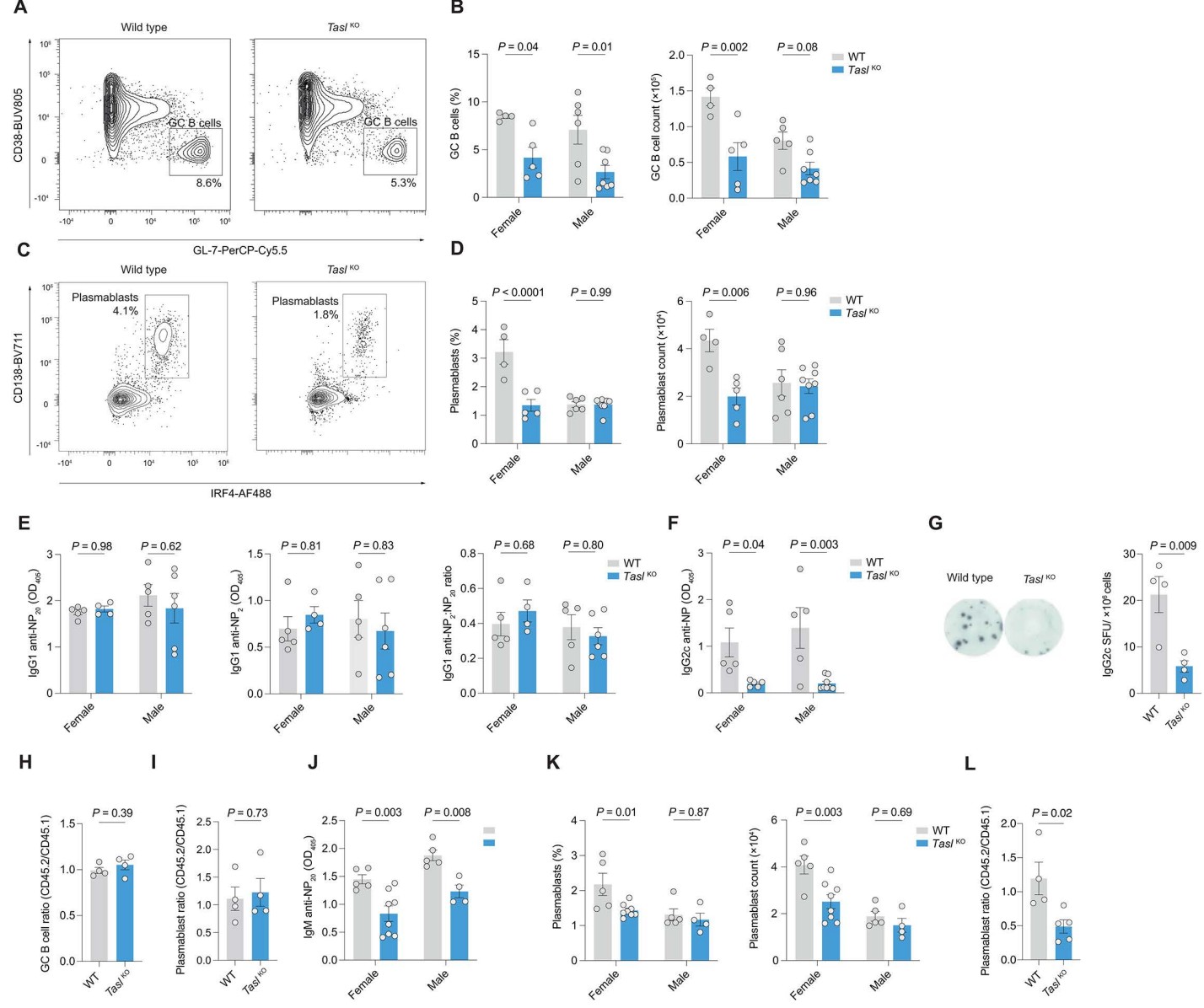

**Fig 2. Loss of TASL impairs humoral immunity. A.** Representative flow cytometric gating of CD19+IgD−CD38−GL-7+ germinal center (GC) B cells from spleens of WT and *Tasl* KO mice at day 14 after immunization with NP-CGG. **B.** Quantification of GC B cell (as in A) proportion and count from spleens of female WT (*n*=4) and *Tasl* KO (*n*=5) mice and male WT (*n*=6) and *Tasl* KO (*n*=7) mice at day 14 after immunization with NP-CGG. Data representative of and pooled from 3 independent experiments. **C.** Representative flow cytometric gating of CD19+IgD−CD138+IRF4+ plasmablasts from spleens of WT and *Tasl* KO mice 14 days after immunization with NP-CGG. **D.** Quantification of plasmablast (as in C) proportion and count from spleens of female WT (*n*=4) and *Tasl* KO (*n*=5) mice, and male WT (*n*=6) and *Tasl* KO (*n*=8) mice. Data representative of and pooled from 3 independent experiments. **E.** ELISA quantification of serum IgG1 anti-NP binding to NP$_{>20}$-BSA and NP$_2$-BSA, and their ratios. Data representative of and pooled from 3 independent experiments. **F.** ELISA quantification of serum anti-NP$_{20}$ IgG2c in female WT (*n*=5) and *Tasl* KO mice (*n*=4), and male WT (*n*=5) and *Tasl* KO (*n*=6) mice, 14 days after immunization with NP-CGG. Data representative of and pooled from 3 independent experiments. **G.** Representative image of ELISpot wells showing IgG2c-producing B cells isolated from WT (*n*=4) and *Tasl* KO (*n*=4) male mice 14 days after immunization with NP-CGG, and quantification of spots. Representative of one independent experiment. **H.** Ratio of splenic CD45.2+ WT or *Tasl* KO GC B cells, normalized to CD45.1+ WT GC B cells in 50:50 competitive bone marrow chimeras 7 days after immunization with sheep red blood cells (*n*=4 female mice per group). Representative of one independent experiment. **I.** Ratio of splenic CD45.2+ WT or *Tasl* KO CD19+IgD−CD138+IRF4+ plasmablasts as in G. **J.** ELISA quantification of anti-NP IgM detected by binding to NP$_{20}$-BSA in sera of female WT (*n*=5) and *Tasl* KO mice (*n*=8), and male WT (*n*=5) and *Tasl* KO (*n*=4) mice, 7 days after immunization with NP-Ficoll. Data representative of and pooled from 2 independent experiments. **K.** Quantification of plasmablast proportion and count in spleens

of female WT ($n=5$) and $Tasl^{KO}$ mice ($n=8$), and male WT ($n=5$) and $Tasl^{KO}$ ($n=4$) mice, 7 days after immunization with NP-Ficoll. Data representative of and pooled from 2 independent experiments. **L**. Ratio of splenic CD45.2+ WT or $Tasl^{KO}$ CD19+IgD−CD138+IRF4+ plasmablasts, normalized to CD45.1+ WT plasmablasts in 50:50 competitive bone marrow chimeras 7 days after immunization with NP-Ficoll ($n=4$ female mice per group). Representative of and pooled from two independent experiments. Statistical significance was calculated using two-way ANOVA Šidák's multiple testing correction (B–F, J–K). or two-tailed unpaired $t$ test (G–I, L). Data are presented as mean±S.E.M. Each point represents a single mouse. The data underlying this figure can be found at: DOI https://doi.org/10.5281/zenodo.18649676.

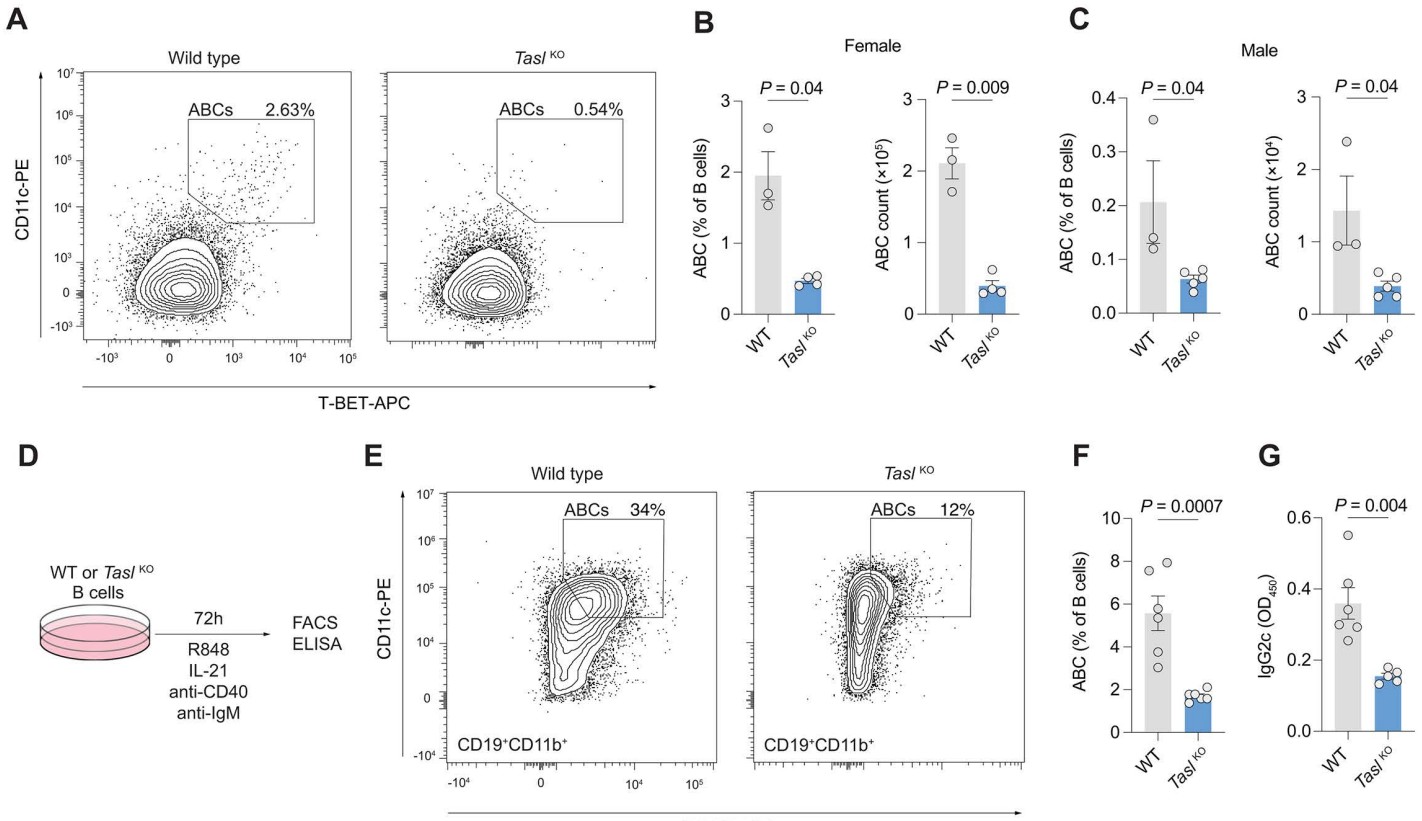

**Fig 3. TASL is required for the formation of age-associated B cells. A.** Representative gating of CD11b+CD11c+T-bet+ age-associated B cells (ABCs) in spleens of 8-month old mice. **B**. Quantification of ABC (as in A) proportion and count in female WT ($n=3$) and $Tasl^{KO}$ ($n=4$) mice. Data pooled from 2 independent experiments. **C**. Quantification of ABC (as in A) proportion and count in male WT ($n=3$) and $Tasl^{KO}$ ($n=5$) mice. Data pooled from 2 independent experiments. **D**. Schematic of ABC in vitro differentiation. ABCs were generated from splenic B cells stimulated with agonistic anti-CD40, anti-IgM F(ab')$_2$, IL-21, and IMQ for 3 days **E**. Representative flow cytometric gating of CD11b+CD11c+T-Bet+ ABCs generated as in D. **F**. Proportion of ABCs generated from male WT and $Tasl^{KO}$ splenic B cells as in D–E ($n=6$ per group). Data pooled from 2 independent experiments. **G**. ELISA quantification of IgG2c antibodies in culture supernatant as in F. Statistical significance was calculated using two-tailed unpaired $t$ test. Data are presented as mean±S.E.M. Each point represents a single mouse. The data underlying this figure can be found at: DOI https://doi.org/10.5281/zenodo.18649676.

## Discussion

In summary, we have shown that in B cells, TASL is required for the effective transduction of signals through endolyso-somal TLRs, generation of predominantly extrafollicular responses, formation of age-related B cells, and development of lupus in a genetically-determined model.

A striking finding across our in vivo experiments was frequent phenotypic sexual dimorphism. The production of plasmablasts in response to both T-dependent and T-independent immunization was significantly decreased in female

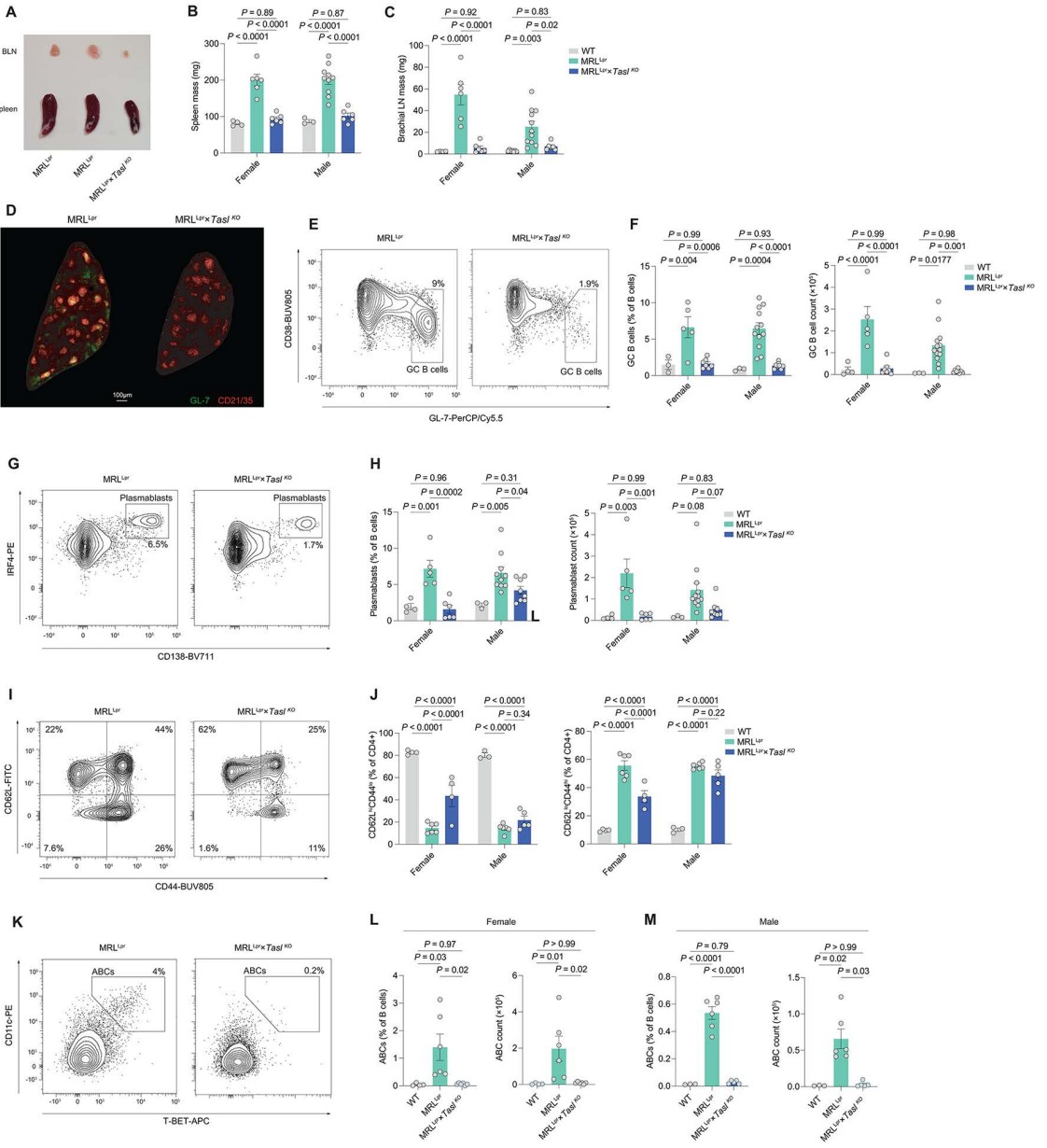

**Fig 4. Absence of TASL prevents the development of lupus in B6.MRL^lpr mice. A.** Representative examples of relative sizes of brachial lymph nodes (BLN) and spleens from 20-week old MRL^Lpr and MRL^Lpr × *Tasl* ^KO **female mice. B.** Quantification of spleen mass in 20-week old female WT (*n*=4), MRL^Lpr (*n*=6), and MRL^Lpr ×*Tasl* ^KO (*n*=6) mice, and male WT (*n*=3), MRL^Lpr (*n*=10), and MRL^Lpr ×*Tasl* ^KO (*n*=6) mice. Data representative of and pooled from 3 independent experiments. **C.** Quantification of BLN mass in 20-week old female WT (*n*=4), MRL^Lpr (*n*=6), and MRL^Lpr ×*Tasl* ^KO (*n*=6) mice, and male WT (*n*=7), MRL^Lpr (*n*=11), and MRL^Lpr ×*Tasl* ^KO (*n*=6) mice. Data representative of and pooled from 3 independent experiments. **D.** Representative splenic GC immunofluorescence images from 20-week old MRL^Lpr and MRL^Lpr ×*Tasl* ^KO female mice labeled with GL-7 and CD21/35 to highlight the GCs and B cell follicles. **E.** Representative flow cytometric gating of CD19⁺IgD⁻CD38⁻GL-7⁺ GC B cells in MRL^Lpr and MRL^Lpr ×*Tasl* ^KO mice. **F.** Quantification of GC B cell proportion and count in 20-week old female WT (*n*=4), MRL^Lpr (*n*=5), and MRL^Lpr ×*Tasl* ^KO (*n*=7) mice, and male WT (*n*=3), MRL^Lpr (*n*=12), and MRL^Lpr ×*Tasl* ^KO (*n*=8) mice. Data representative of and pooled from 3 independent experiments. **G.** Representative flow cytometric gating of CD19⁺IgD⁻CD138⁺IRF4⁺ plasmablasts in MRL^Lpr and MRL^Lpr ×*Tasl* ^KO mice. **H.** Quantification of plasmablast proportion and count in 20-week old female WT (*n*=4), MRL^Lpr (*n*=5), and MRL^Lpr ×*Tasl* ^KO (*n*=6) mice, and male WT (*n*=3), MRL^Lpr (*n*=12), and MRL^Lpr ×*Tasl* ^KO (*n*=8) mice. Data representative of and pooled from 3 independent experiments. **I.** Representative flow cytometric gating of CD62L and CD44 quadrants of CD4⁺ T cells in spleens of 20-week old MRL^Lpr and MRL^Lpr ×*Tasl* ^KO mice. **J.** Quantification of naïve (CD4⁺CD62L^hiCD44^lo) and effector (CD4⁺CD62L^loCD44^hi) T cells in spleens of 20-week old female WT (*n*=4), MRL^Lpr (*n*=6), and MRL^Lpr ×*Tasl* ^KO (*n*=4) mice, and male WT (*n*=3), MRL^Lpr (*n*=6), and MRL^Lpr ×*Tasl* ^KO (*n*=5) mice. Data

representative of and pooled from 2 independent experiments. **K**. Representative flow cytometric gating of CD19+CD11b+CD11c+T-Bet+ ABCs in spleens of 20-week old MRL^Lpr and MRL^Lpr ×Tasl ^KO mice. **L**. Quantification of ABC proportion and count in spleens of 20-week old female WT ($n=4$), MRL^Lpr ($n=6$), and MRL^Lpr ×Tasl ^KO ($n=6$) mice. Data pooled from 2 independent experiments. Quantification of ABC proportion and count in spleens of 20-week old male WT ($n=3$), MRL^Lpr ($n=6$), and MRL^Lpr ×Tasl ^KO ($n=5$) mice. Data pooled from of 2 independent experiments. Statistical significance was calculated using two-way ANOVA with Šidák's multiple testing correction. Data are presented as mean±S.E.M. Each point represents a single mouse. The data underlying this figure can be found at: DOI https://doi.org/10.5281/zenodo.18649676.

mice, and whilst trending in the same direction in males, the reduction was far less. In addition, the reduction in frequency of plasmablasts in MRL^Lpr × Tasl ^KO female mice was greater than that seen in male mice. These observations mirror those of sexually dimorphic *TASL* expression in humans. It was previously found that the lupus risk SNP rs887369 only acted as an expression quantitative trait locus in females, in which it led to higher *TASL* expression [2]. This effect was magnified following immune cell stimulation. *TASL* frequently exhibits loss of X-chromosome inactivation (XCI), and the extent to which *TASL* interacts with other XCI-escaping genes across the dynamic immune response remains to be determined.

We made the observation that loss of TASL predominantly affects the extrafollicular B cell response. Whilst GC B cell numbers were reduced in *Tasl* ^KO mice, we did not detect differences in antigen-specific IgG1 or defective antibody affinity maturation, nor was there evidence of a competitive disadvantage in a bone marrow chimera setting. Signaling through TLR7 and TLR9 modulates the relative balance of GC and extrafollicular responses depending on context [26], with downstream MyD88 required for IgG2a/c class switching and T-bet expression [27]. TLR7 signaling is a critical element of the pathogenesis of SLE, and also required for ABC formation [22]. ABCs are associated with systemic autoimmunity in a number of diseases, and their link with SLE is especially well established. Collectively these findings reinforce the previously established function of TASL as an adaptor for IRF5. There were phenotypic differences to *Irf5*-/- mice however, with an increase in T2 transitional B cells not seen in IRF5 deficiency [28–30]. Another difference compared with *Irf5*-/- B cells or *IRF5* knockdown in human B cells is in activation following stimulation [31,32]. We noted a paradoxical increase in CD86 expression following stimulation with CpG in vitro, despite an overall reduction in CD69 and downregulation of IgD, and which is also reflected in our RNA sequencing data. Again, it is not immediately clear why this occurs, but may reflect differences in tonic signaling. We also observed a sex-specific reduction in resting IgG levels, not seen in equivalent reports of *Irf5* deletion [33]. Why these differences occur requires further investigation.

An outstanding question remains around the cell-specific functions of TASL. Since *Tasl* is a single exon gene, generation of a LoxP-flanked allele to allow conditional deletion is technically challenging. It therefore remains unclear whether its effect on pDC or B cell signaling is dominant in disease or infection models.

The location of TASL in the TLR7-IRF5 signaling pathway makes it a potentially attractive therapeutic target for SLE and other related autoimmune disease. A molecule targeting SLC15A4 that leads to the degradation of TASL in vitro has recently been identified, which warrants further characterization [34].

## Methods

### Ethics statement

All procedures and experiments were performed in accordance with the UK Scientific Procedures Act (1986) under a project license authorized by the UK Home Office (PPL no. PP1971784).

### Mice

B6.MRL-*Fas^lpr*/J (strain 000482) were purchased from the Jackson Laboratory. C57BL/6J-Tasl^em3H/H mice were generated by the Mary Lyon Centre at MRC Harwell, via CRISPR/Cas9-induced mutation deleting 10 nucleotides in exon ENS-MUSE00000697605 of the *Tasl* gene. B6.SJL.CD45.1 mice were obtained from the central breeding facility of the University of Oxford.

Males and females between the ages of 8–16 weeks were used, except where stated separately for individual experiments. Mice were bred and maintained under specific pathogen-free conditions at the Kennedy Institute of Rheumatology, University of Oxford.

## Immunization

50µg NP$_{(30–39)}$-CGG (Biosearch Tech) or NP-Ficoll was mixed with alum (Thermo Fisher Scientific) at a 1:1 ratio and rotated at room temperature for 30 min, before being injected intraperitoneally. For NP-CGG and NP-Ficoll immunisations, day 14 and day 7 were used as timepoints respectively, unless stated otherwise.

For SRBC immunization, 1 ml of sterile SRBCs in Alsever's solution (Thermo Fisher Scientific) was washed twice with 10 ml ice-cold PBS, then reconstituted in 2–3 ml of PBS. 200 µl was then injected intraperitoneally.

## Bone marrow chimera generation

Recipient female B6.SJL.CD45.1 mice received two doses of 5.5Gy irradiation 4 hours apart, and were then injected intravenously with 4 × 10$^6$ bone marrow (BM cells) in 200µl PBS, made up of 50:50 CD45.2$^+$ *Tasl*$^{KO}$ and CD45.1$^+$ WT, or CD45.2$^+$ WT and CD45.1$^+$ WT BM cells from sex- and age-matched donor mice. Recipient mice were given antibiotics (Baytril, Bayer corporation) in their drinking water for two weeks. Eleven weeks after BM reconstitution, recipient mice were immunized with either SRBC or NP-Ficoll. Spleens were analyzed 9 days post-immunization with SRBCs and 7 days post-immunization with NP-Ficoll.

## Flow cytometry

Spleens were mashed through a 70 µm strainer (VWR). Red blood cells were lysed with ACK Lysis buffer (Gibco). Single-cell suspensions were incubated with Fixable Viability Dye eFluor 780 (eBioscience) or Zombie UV Fixable Viability Dye (BioLegend) in PBS for 30 min on ice, then Fc block (BD Biosciences) for 5 mins, followed by surface antibodies in a 50:50 solution of FACS buffer and Brilliant Stain buffer (BD Biosciences) for 30 mins on ice. Cells were fixed in freshly prepared 4% PFA for 10 mins at room temperature.

For intracellular staining, cells were fixed using FoxP3/Transcription Factor Fixation/Permeabilization Concentrate and Diluent (eBioscience) for 30 mins at room temperature, then permeabilised in Permeabilization buffer (eBioscience). Cells were then incubated with intracellular antibodies diluted in permeabilization buffer for 45 mins at room temp.

The following antibodies were used (Table 1):

Cells were resuspended in FACS buffer and acquired on either a BD Fortessa X20 or Cytek Aurora flow cytometer.

## Cell isolation and culture

Splenic B cells were isolated by using the Pan B cell Isolation II Kit (Milteny Biotec), with resulting purity of over 90%. For proliferation analysis, cells were incubated with Cell Trace Violet (Life Technologies) in PBS for 20 mins at room temperature, before adding complete media and washing. Isolated B cells were then cultured in RPMI-1640 medium (Gibco) with 10% FCS (Gibco), 50 µM 2-mercaptoethanol (Gibco), 1 mM glutamine (Gibco), 1 mM sodium pyruvate (Gibco), 10 mM HEPES buffer (Gibco), and 100 IU/ml penicillin and streptomycin (Life Technologies Ltd).

For TLR stimulation, either 100 nM CpG-B (ODN1826) (Miltenyi Biotec), 5 µg/ml IMQ (Invitrogen), or 10 µg/ml LPS (Merck) was used, for times indicated in figures.

For ABC in vitro culture, a differentiation cocktail of 0.5 µg/ml IMQ, 1 µg/ml anti-CD40 (clone FGK45.4, Miltenyi Biotec), 1 µg/ml anti-IgM F(ab')$_2$ (Jackson ImmunoResearch), and 50ng/ml IL-21 (PeproTech) was used.

## iGB culture system

The iGB culture system was used as previously described [18,35]. The 40LB fibroblast cell line was cultured in high-glucose DMEM with Glutamax (Thermofisher) medium containing 10% FCS (Gibco) and 50 IU/ml penicillin and

**Table 1. Antibodies.**

| Antibody | Fluorochrome | Clone | Dilution (1/n) | Manufacturer | Cat no: |
|---|---|---|---|---|---|
| anti-B220 | BV510 | RA3-6B2 | 100 | BioLegend | 103248 |
| anti-CD19 | BV421 | 6D5 | 100 | BioLegend | 115538 |
| anti-CD19 | BV510 | 6D5 | 100 | BioLegend | 115546 |
| anti-CD19 | BV650 | 6D5 | 100 | BioLegend | 115541 |
| anti-CD19 | BV785 | 6D5 | 100 | BioLegend | 115543 |
| anti-CD21/35 | BV711 | 7E9 | 200 | BioLegend | 123435 |
| Anti-CD21/35 | AF647 | 7E9 | 100 | BioLegend | 123423 |
| anti-CD24 | BV711 | M1/69 | 200 | BD Bioscience | 563371 |
| anti-BP-1 (CD249) | BUV737 | 6C3 | 100 | BD Bioscience | 741743 |
| anti-IgD | BV650 | 11-26c.2a | 400 | BioLegend | 405721 |
| anti-IgD | AF700 | 11-26c.2a | 400 | BioLegend | 405730 |
| anti-CD43 | PE | S11 | 200 | BioLegend | 143205 |
| anti-CD23 | BV786 | BSB4 | 150 | BD Bioscience | 563988 |
| anti-IgM | BUV395 | II/41 | 150 | BD Bioscience | 743329 |
| anti-CD69 | BV711 | H1.2F3 | 200 | BioLegend | 104537 |
| anti-CD86 | BV785 | GL-1 | 100 | BioLegend | 105043 |
| anti-CD138 | BV711 | 281−2 | 250 | BioLegend | 142519 |
| anti-CD4 | BV605 | RM4–5 | 150 | BioLegend | 100547 |
| anti-CD8α | PE/Cy7 | 53-6.7 | 100 | BioLegend | 100721 |
| anti-CD8α | BV711 | 53-6.7 | 200 | BioLegend | 100747 |
| anti-CD38 | BUV805 | 90/CD38 | 400 | BD Bioscience | 741955 |
| anti-CXCR4 | BUV737 | 2B11/CXCR4 | 50 | BD Bioscience | 741783 |
| anti-CXCR5 | BV421 | L138D7 | 50 | BioLegend | 145511 |
| anti-GL7 | Pacific Blue | GL7 | 100 | BioLegend | 144614 |
| anti-GL7 | PerCP/Cy5.5 | GL7 | 200 | BioLegend | 144609 |
| anti-GL7 | AF488 | GL7 | 50 | BioLegend | 144611 |
| anti-TCRβ | BV510 | H57-597 | 100 | BioLegend | 109233 |
| anti-CD11c | PE | N418 | 200 | BioLegend | 117307 |
| anti-CD11c | BV650 | N418 | 100 | BioLegend | 117339 |
| anti-CD11b | PerCP/Cy5.5 | M1/70 | 200 | BioLegend | 101228 |
| anti-CD62L | FITC | MEL-14 | 100 | BioLegend | 104405 |
| anti-CD44 | BUV805 | IM7 | 200 | BD Bioscience | 741921 |
| anti-I-A/I-E | AF647 | M5/114.15.2 | 500 | BioLegend | 107618 |
| anti-PDCA1 | BV421 | 927 | 100 | BioLegend | 127023 |
| anti-F4/80 | FITC | BM8 | 100 | BioLegend | 123107 |
| anti-PD1 | FITC | 29F.1A12 | 100 | BioLegend | 135213 |
| **Intracellular** | | | | | |
| anti-IRF4 | AF488 | 3E4 | 100 | BioLegend | 646406 |
| anti-IRF4 | PE | 3E4 | 100 | Invitrogen | 12-9858-82 |
| anti-Tbet | APC | 4B10 | 100 | BioLegend | 644814 |
| anti-FoxP3 | PE/Cy7 | FJK-16s | 100 | Invitrogen | 25-5773-80 |
| anti-IL-6 | PE | MP5-20F3 | 100 | BioLegend | 504503 |
| anti-TNFα | FITC | MP6-XT22 | 100 | BioLegend | 506304 |
| anti-IFNγ | PE | XMG1.2 | 100 | BioLegend | 505808 |
| anti-BCL6 | AF647 | K112-91 | 100 | BD Bioscience | 561525 |

streptomycin (Life Technologies Ltd) in T75 tissue culture flasks (Sarstedt). When cells were confluent, they were detached using trypsin/EDTA (Gibco), washed, then resuspended in 5 ml medium in 15 ml tubes and irradiated with a dose of 80 Gy. Irradiated cells were then washed, and seeded at $0.5 \times 10^6$ in 6-well culture plates (Falcon), then incubated overnight at 37°C and 5% $CO_2$.

Naïve B cells were isolated using anti-CD43 microbeads (Miltenyi) and added to wells with irradiated 40 LBs cells for 5 days in complete RPMI medium, as described above, with 1 ng/ml IL-4 (Peprotech). Cells were then collected and analyzed by flow cytometry. iGCBCs were defined as $CD19^+IgD^{lo}CD38^{lo}GL-7^+$.

## ELISA

96-well plates were coated with either $NP_2$ or $NP_{>20}$ conjugated to BSA, at 5 µg/ml in bicarbonate/carbonate coating buffer at 4 °C overnight. Plates were washed in PBS, then blocked with 5% skimmed milk in PBS for 2 hour at 37 °C. After washing with PBS, serum from immunized mice was diluted in 1% skimmed milk and then added to plates and incubated for 1 hour at 37 °C. Plates were then washed multiple times with PBS-Tween20 (0.05%) (PBS-T), before alkaline phosphatase-conjugated goat anti-mouse IgG, IgG1, IgG2c, or IgM detection antibodies (SouthernBiotech) were added and incubated at 37 °C for 1 hour. After washing, alkaline phosphatase substrate (Sigma) was added to plates and left to develop for 5 min at room temperature, after which readings were taken every 5 mins on a FLUOstar Omega plate reader.

For Ig isotyping, Mouse Ig Isotyping Uncoated ELISA kit (Thermo Fisher Scientific) was used according to manufacturer's protocol.

For detection of auto-antibodies, 96-well plates were coated with UltraPure Salmon Sperm DNA Solution (Invitrogen) at 100 µg/ml in PBS at 4 °C overnight for anti-dsDNA antibodies, or 1 µg/ml Sm/RNP (Arotec Diagnostics) in PBS at 4 °C overnight for anti-Sm/RNP antibodies. Plates were washed in PBS, then blocked with 2% BSA for 1 hour at 37 °C. Plates were then washed in PBS, and mouse serum diluted in PBS containing 1% BSA was added and incubated for 2 hour at 37 °C. After multiple washes in PBS-T, alkaline phosphatase-conjugated goat anti-mouse IgG, IgG2c, or IgM detection antibodies (SouthernBiotech) were then added and incubated at 37 °C for 1 hour. Plates were then washed with PBS-T, then alkaline phosphatase substrate (Sigma) was added to plates and left to incubate for 5 mins at room temperature. Readings were then taken every 3 mins on a FLUOstar Omega plate reader at 405 nm.

## ELISpot assay

Low fluorescent PVDF Fluorospot plates (Mabtech) were coated with anti-IgG2c (Southern Biotech) overnight at 4 °C. Plates were washed with PBS and then blocked with PBS containing 1% gelatin. After washing with PBS, single-cell splenocyte suspensions from mice 14 days after immunization with NP-CGG were added to plates in complete RPMI and incubated overnight at 37 °C. Plates were then washed with distilled water-0.1% Tween and PBS-0.05% Tween. Alkaline phosphatase-conjugated goat anti-mouse IgG (Southern Biotech) was then added and incubated for 1 hour at 37 °C. Plates were then washed with PBS-0.05% Tween and AMP/BCIP substrate solution overnight at 4 °C. Plates were then washed with distilled water and left to dry, before being analyzed on an AID ELISpot reader.

## Immunohistochemistry

Spleens were fixed overnight at 4 °C in Antigenfix (DiaPath), then washed in PBS and left in 30% sucrose in PBS overnight at 4 °C. Spleens were then snap frozen in methanol on dry ice and stored at −70 °C. Spleens were sectioned at 8−12 µm thickness and rehydrated in PBS at room temperature, before being permeabilised and blocked in PBS containing 10% goat serum and 0.1% Tween-20. Spleens were then stained overnight at 4 °C with anti-GL-7 AF488 (Biolegend) and anti-CD21/35 AF647 (Biolegend). Slides were then washed and mounted before being imaged with an Airyscan 2 and analyzed in ImageJ.

## Anti-nuclear antibody imaging

Mouse serum was added to Kallestad Hep-2 slides (BioRad) and incubated for 30 mins at room temperature in the dark. Slides were washed in PBS, then AF488-conjugated goat anti-mouse IgG (Jackson ImmunoResearch) was added to slides and incubated for 30 mins at room temperature in the dark. Slides were then washed and mounted, before being imaged with a ZEISS LSM 980 and analyzed in ImageJ.

## Western blotting

Cells were lysed in RIPA buffer (Merck) supplemented with phosphatase and protease inhibitors (cOmplete ULTRA Tablets and PhosSTOP, Roche) for 40 mins with regular vortexing, before being centrifuged at 15,000 $g$ for 15 mins at 4 °C. The supernatant was then collected. Protein was quantified with a Pierce BCA Protein Assay kit (Thermo Fisher Scientific). Samples were denatured in Laemmli buffer (BioRad) containing 10% 2-mercaptoethanol for 5 mins at 95 °C, then cooled immediately on ice. Samples were then run on a 10% gel (Mini-PROTEAN TGX Precast Gels, BioRad).

Protein was then transferred to a PVDF membrane (BioRad). Membranes were blocked in 2.5% skimmed milk in TBS for 1h, washed with TBS-Tween20 (0.05%) (TBS-T), then incubated overnight at 4 °C with primary antibodies anti-TASL (1:1000) (Atlas) and anti-vinculin (1:1000) (Cell Signaling Technology) diluted in TBS-T containing 2.5% BSA. Membranes were then washed with TBS-T before adding HRP-conjugated goat anti-rabbit secondary antibody (Cell Signaling Technology), and incubating for 1 hour at room temperature. SuperSignal West Pico PLUS Chemiluminescent Substrate (ThermoFisher) was then added to membranes, and chemiluminescence was used to detect signal.

## Quantitative PCR

Cells were lysed in RLT Plus (Qiagen) containing 1% 2-mercaptoethanol. RNA was isolated using an RNeasy Plus Mini kit (Qiagen), according to manufacturer's instructions. Isolated RNA was reverse transcribed to cDNA using the High Capacity RNA-to-cDNA kit (Thermo Fisher).

qPCR reaction was performed using TaqMan Fast Advanced Master Mix (Applied Biosystems) and TaqMan primers for *Isg15* (Mm01705338_s1), *Il6* (Mm00446190_m1), *Il10* (Mm01288386_m1), *Irf7* (Mm00516793_g1), *Mx1* (Mm00487796_m1), *Prdm1* (Mm00476128_m1), and *Ubc* (Mm00476128_m1). Data was acquired on an Applied Biosystems Viia7 System, and expression was normalized to *Ubc* expression using the ΔΔCT method.

## Bulk RNA sequencing analysis

Isolated splenic B cells were stimulated for 24 hour with 100 nM CpG. Cells were then lysed in RLT plus containing 1% 2-mercaptoethanol, and RNA was isolated using RNeasy Plus Mini kit (Qiagen). Next generation paired-end sequencing was performed by Azenta using the Illumina NovaSeq 6000 platform, with 20 M reads taken per sample. Transcripts were counted using *Salmon*.

After standard processing, data was filtered to genes that had expression higher than 1 CPM in at least 50% of samples. Normalization and differential expression analysis were then performed with edgeR.

For gene set enrichment analysis (GSEA), the ClusterProfiler package was used with gene ontology terms related to biological processes (GO BP). The Benjamini-Hochberg method was used for $P$-value adjustment to control for false discovery rate.

## Supporting information

**S1 Fig. A**. Immunoblot of TASL in WT and *Tasl* [KO] splenocytes. Vinculin is used as a loading control. Representative of 2 independent experiments. **B**. Flow cytometric quantification of frequency of live CD19+ B cells in spleens of WT ($n$ = 10) and *Tasl* [KO] ($n$ = 8) mice. Data representative of and pooled from 3 independent experiments. **C**. Count of live

CD19$^+$ B cells in spleens of WT ($n = 5$) and *Tasl* $^{KO}$ ($n = 5$) mice. Data representative of and pooled from 2 independent experiments. **D**. Representative gating and quantification of frequency of CD19$^+$CD23$^+$CD21$^{int}$ follicular B cells and CD19$^+$CD23$^-$CD21$^+$ marginal zone (MZ) B cells in spleens of WT ($n = 12$) and *Tasl* $^{KO}$ ($n = 8$) B cells. Data representative of and pooled from 3 independent experiments. **E**. Count of follicular B cells and MZ B cells, gated as in D, in spleens of WT ($n = 7$) and *Tasl* $^{KO}$ ($n = 5$) mice. Data representative and pooled from 2 independent experiments. **F**. Representative gating and quantification of frequency of CD19$^+$CD93$^+$ T1 (CD23$^-$IgM$^+$), T2 (CD23$^+$IgM$^+$), and T3 (CD23$^+$IgM$^-$) B cells in spleens of WT ($n = 12$) and *Tasl* $^{KO}$ ($n = 8$) mice. Data representative of and pooled from 3 independent experiments. **G**. Count of transitional B cell populations, gated as in F, in spleens of WT ($n = 7$) and *Tasl* $^{KO}$ ($n = 5$) mice. Data representative of and pooled from 2 independent experiments. **H**. Quantification of total TCRβ$^+$ T cell frequency in spleens of WT ($n = 10$) and *Tasl* $^{KO}$ ($n = 7$) mice. Data representative of and pooled from 3 independent experiments. **I**. Quantification of total TCRβ$^+$ T cell count in spleens of WT ($n = 7$) and *Tasl* $^{KO}$ ($n = 5$) mice. Data representative of and pooled from 2 independent experiments. **J**. Representative gating and quantification of frequency and count of splenic CD11b$^+$F4/80$^+$ macrophages in spleens of WT ($n = 12$) and *Tasl* $^{KO}$ ($n = 8$) mice. Data representative of and pooled from 3 independent experiments. **K**. Count of macrophages, as gated as in F, in spleens of WT ($n = 7$) and *Tasl* $^{KO}$ ($n = 5$) mice. Data representative of and pooled from 2 independent experiments. **L**. Representative gating and quantification of frequency of plasmacytoid dendritic cells (pDCs) in spleens of WT ($n = 12$) and *Tasl* $^{KO}$ ($n = 8$) mice. Data representative and pooled from 3 independent experiments. **M**. Count of pDCs, gated as in F, in spleens of WT ($n = 7$) and *Tasl* $^{KO}$ ($n = 5$) mice. Data representative of and pooled from 2 independent experiments. **N**. Representative gating and quantification of frequency and count of for IgM$^-$IgD$^-$B220$^+$ pre-pro-, pro- and pre- B cells in the bone marrow of WT ($n = 4$) and *Tasl* $^{KO}$ ($n = 4$) mice. Equivalent Hardy fractions are shown in parentheses. Data representative of 2 independent experiments. **O**. Representative gating and quantification of frequency and count of B220$^+$IgD$^-$IgM$^+$ immature B cells and B220$^+$IgD$^+$ mature B cells in the bone marrow of WT ($n = 4$) and *Tasl* $^{KO}$ ($n = 4$) mice. Equivalent Hardy fractions are shown in parentheses. Data representative of 2 independent experiments. Statistical significance was calculated using two-tailed unpaired *t* test (B–E, H–M) or two-way ANOVA with Šidák's multiple testing correction (F, G, N, and O). Data are presented as mean ± S.E.M. Each point represents a single mouse. The data underlying this figure can be found at: DOI https://doi.org/10.5281/zenodo.18649676
(PDF)

**S2 Fig. A**. Representative histogram and proliferation index of isolated splenic B cells from WT or *Tasl* $^{KO}$ mice ($n = 4$) stimulated for 72 hour with either lipopolysaccharide (LPS), imiquimod (IMQ), or CpG. Data representative of 2 independent experiments. **B**. Flow cytometric measurement of CD69 gMFI in isolated splenic B cells from either WT ($n = 4$) or *Tasl* $^{KO}$ ($n = 4$) male mice after stimulation with either LPS, IMQ, or CpG for 24 hour. Representative of two independent experiments. **C**. Flow cytometric measurement of CD86 gMFI in isolated splenic B cells from either WT ($n = 4$) or *Tasl* $^{KO}$ ($n = 4$) male mice after stimulation with either LPS, IMQ, or CpG for 24 hour. Representative of two independent experiments. **D**. Quantification of IgD$^+$ B cells as percentage of total B cells, after stimulation of isolated splenic B cells from male WT ($n = 4$) or *Tasl* $^{KO}$ ($n = 4$) mice with either LPS, IMQ, or CpG for 72 hour. Representative of 2 independent experiments. **E**. Quantification of plasmablasts as CD19$^+$IgD$^-$CD138$^+$IRF4$^+$ B cells after culture of male B cells with either LPS, IMQ, or CpG for 72 hour ($n = 4$). Representative of 2 independent experiments. Statistical significance was calculated using two-tailed unpaired *t* test (A, **G**) or two-way ANOVA with Šidák's multiple testing correction (B–E). Data are presented as mean ± S.E.M. Each point represents a single mouse. The data underlying this figure can be found at: DOI https://doi.org/10.5281/zenodo.18649676
(PDF)

**S3 Fig. A**. Immunoglobulin isotyping of unimmunized female WT ($n = 5$) or *Tasl* $^{KO}$ ($n = 3$) or male WT ($n = 4$) or *Tasl* $^{KO}$ ($n = 4$) mice. Data are representative of 2 independent experiments. Statistical significance calculated by two-way ANOVA

with Šidák's multiple testing correction. Data are presented as mean ± S.E.M. Each point represents a single mouse. The data underlying this figure can be found at: DOI https://doi.org/10.5281/zenodo.18649676
(PDF)

**S4 Fig. A**. Schematic of iGB culture system. Isolated naïve splenic B cells from WT ($n=7$) or *Tasl* KO ($n=7$) mice were seeded on a layer of irradiated fibroblasts expressing BAFF and CD40L, with IL-4 and incubated for 5 days. **B**. Quantification of CD38⁻GL-7⁺ iGCB cells from (**F**). Statistical significance was calculated using two-tailed unpaired *t* test. Data are presented as mean ± S.E.M. Each point represents a single mouse. The data underlying this figure can be found at: DOI https://doi.org/10.5281/zenodo.18649676S
(PDF)

**S5 Fig. A**. Representative flow cytometric gating of CD19⁻TCRb⁻CD11c⁺pDCA1⁺ pDCs in MRL^Lpr and MRL^Lpr × *Tasl* KO mice. **B**. Quantification of pDC proportion and count in 20-week old female WT ($n=4$), MRL^Lpr ($n=6$), and MRL^Lpr × *Tasl* KO ($n=5$) mice, and male WT ($n=3$), MRL^Lpr ($n=4$), and MRL^Lpr × *Tasl* KO ($n=5$) mice. Data representative of and pooled from 2 independent experiments. **C**. Quantification of CD86^hi pDCs in 20-week old female WT ($n=4$), MRL^Lpr ($n=6$), and MRL^Lpr × *Tasl* KO ($n=5$) mice, and male WT ($n=3$), MRL^Lpr ($n=4$), and MRL^Lpr × *Tasl* KO ($n=5$) mice. Data representative of and pooled from 2 independent experiments. **D**. Representative images of anti-nuclear antibodies (ANAs) in serum of 20-week old WT, MRL^Lpr and MRL^Lpr × *Tasl* KO mice, detected by immunofluorescence. Scale bar represents 100µm. **E**. ELISA quantification of anti-dsDNA IgG2c in sera of 20-week old female WT ($n=4$), MRL^Lpr ($n=3$), and MRL^Lpr × *Tasl* KO ($n=7$) mice, and male WT ($n=3$), MRL^Lpr ($n=10$), and MRL^Lpr × *Tasl* KO ($n=7$) mice. Data representative of and pooled from 3 independent experiments. **F**. ELISA quantification of anti-Sm/RNP IgG2c in sera of 20-week old female WT ($n=4$), MRL^Lpr ($n=3$), and MRL^Lpr × *Tasl* KO ($n=9$) mice, and male WT ($n=3$), MRL^Lpr ($n=10$), and MRL^Lpr × *Tasl* KO ($n=7$) mice. Data representative of and pooled from 3 independent experiments. Statistical significance was calculated using two-way ANOVA with Šidák's multiple testing correction. Data are presented as mean ± S.E.M. Each point represents a single mouse. The data underlying this figure can be found at: DOI https://doi.org/10.5281/zenodo.18649676
(PDF)

**S1 Raw Images. Unedited original western blots used in this study.**
(PDF)

## Acknowledgments

We thank J. Bonifacio Lopes and M. Borsa for their assistance and helpful guidance, and the Kennedy Institute BSU staff for their support. The MLC generated the C57BL/6J-Tasl^em3H/H mouse strain as part of its commitment to the Genome Editing Mice for Medicine project funded [MC_UP_1502/3] by the Medical Research Council. The research reported in this publication is solely the responsibility of the authors and does not necessarily represent the official views of the Medical Research Council.

## Author contributions

**Conceptualization:** Julia C. Johnstone, Timothy J. Vyse, Alex J. Clarke.

**Formal analysis:** Julia C. Johnstone, Robert Mitchell, Alex J. Clarke.

**Investigation:** Julia C. Johnstone, Robert Mitchell.

**Writing – original draft:** Julia C. Johnstone, Alex J. Clarke.

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
