## [Editor Report · Decision Letter 0]

30 Jul 2025

Dear Dr Clarke,

Thank you for submitting your manuscript entitled "TASL is required for age-related B cell formation and autoimmunity" for consideration as a Research Article by PLOS Biology.

Your manuscript has now been evaluated by the PLOS Biology editorial staff, as well as by an academic editor with relevant expertise, and I am writing to let you know that we would like to send your submission out for external peer review under our anti-scooping protection policy given recent publications.

Once your full submission is complete, your paper will undergo a series of checks in preparation for peer review. After your manuscript has passed the checks it will be sent out for review. To provide the metadata for your submission, please Login to Editorial Manager (https://www.editorialmanager.com/pbiology) within two working days, i.e. by Aug 01 2025 11:59PM.

Kind regards,

Melissa

Melissa Vazquez Hernandez, Ph.D.

Associate Editor

PLOS Biology

---

## [Decision Letter · Decision Letter 1]

2 Sep 2025

Dear Dr Clarke,

Thank you for your patience while your manuscript "TASL is required for age-related B cell formation and autoimmunity" was peer-reviewed at PLOS Biology. It has now been evaluated by the PLOS Biology editors, an Academic Editor with relevant expertise, and by three independent reviewers.

In light of the reviews, which you will find at the end of this email, we would like to invite you to revise the work to thoroughly address the reviewers' reports. As you will see below, majority of reviewers are positive about the relevance of the study, yet some concerns have raised during revision. Reviewer 1 thinks there is not enough evidence to support the role of TASL in extrafollicular vs GC pathways, and suggests measuring antigen-specific GC and plasmablast responses (including in chimeras), testing GC-like differentiation in vitro, and using immunofluorescence to assess GC formation in lupus-prone mice. S/he mentions that to increase novelty the study should investigate the B-cell intrinsic role of TASL using BM chimera approach. Reviewer 2 asks the authors to discuss several points like how does theirp henotype overlap with Irf5ko mice, why effects are strongest downstream of TLR9 but minimal for TLR7 or TLR4, and discrepancies such as unchanged plasmablasts despite reduced IgG2c, reduced steady-state Ig levels, and mismatched flow plots vs bar graphs. S/he would like to know what effect does NP-Ficoll have on the generation of PBs and stronger evidence on their claim that Tasl is required for transduction of endolysosomal signaling. Reviewer 3 requires some clarifications on sex differences in TLR stimulation, labeling male vs female data directly on graphs, merging the two brief sections on mrl.lpr data, and clarifying the sex of representative images in Figure 4a.

IMPORTANT: While the study is being considered under our anti-scooping policy, we think the suggestions made by the reviewers will enhance the strength and novelty of the study, and we would require them for a successful revision.

Given the extent of revision needed, we cannot make a decision about publication until we have seen the revised manuscript and your response to the reviewers' comments. Your revised manuscript is likely to be sent for further evaluation by all or a subset of the reviewers.

**IMPORTANT - SUBMITTING YOUR REVISION**

*Re-submission Checklist*

*Published Peer Review*

*PLOS Data Policy*

*Blot and Gel Data Policy*

Sincerely,

Melissa

Melissa Vazquez Hernandez, Ph.D.

Associate Editor

PLOS Biology

REVIEWERS' COMMENTS

Reviewer #1:

TASL is an adaptor in the IRF5 signaling pathway and is involved in the phosphorylation of IRF5 in response to TLR7 or TLR9 stimulation. TASL has been shown to play a role in the activation and function of B cells, and in the development of autoimmunity in chemically induced models of lupus. The current manuscript by Johnstone et al. delineates the role of TASL in humoral and autoimmune B cell responses in the extrafollicular age-associated B cell (ABC) and follicular GC differentiation pathways in the context of foreign antigen and spontaneous autoimmunity. The manuscript is novel and significant in highlighting the role of TASL in GC and ABC responses in both foreign antigen driven and spontaneous lupus-like autoimmune responses. I have the following comments and addressing the comments will significantly improve the manuscript.

(1) The extrafollicular ABC and follicular GC responses have been implicated in both protection and in autoimmunity. In the current manuscript, the authors used NP-CGG as a model antigen to determine the role of TASL in protective humoral responses via the ABC and GC pathways. The authors found reduced GC and plasmablast (PB) responses in TASL KO mice compared to wild type control mice. They did not find any difference in affinity maturation and anti-NP IgG1 (Th2) response, but anti-NP IgG2c (Th1) response was reduced. They then went on to show no difference in GC response between wild type and TASL KO GC responses in a competitive bone marrow chimera environment. From these data the authors concluded that TASL is primarily required for the extrafollicular response, with a limited role in the GC reaction. In a competitive chimera, was there any difference in plasmablast response as this was not shown? Also, was there any difference in anti-SRBC IgG1 and IgG2c responses? The authors should also measure NP-specific GC and plasmablast responses as total GC and PB responses constitute both immunization-induced and spontaneous GC B cell and PB responses.

(2) In Figure 3A-C, the authors determined spontaneous ABC formation in 8-month-old wild type and TASL KO mice without examining GC formation at this age. In Figure 3D-F, they differentiate B cells into ABCs through in vitro stimulation with R848, IL-21, anti-CD40 and anti-IgM. These stimuli also differentiate B cells into GC-like cells and authors could demonstrate whether GC-like B cell responses were equally reduced in the absence of TASL. Both steady-state GC and ABC responses could generate IgG2c Abs.

(3) Plasmablast, ABC and GC responses are all equally reduced in lupus-prone B6.MRLlpr mice and therefore arguing for ABC alone is not well supported.

(4) Showing immunofluorescent imaging of GC reaction in lupus-prone mice would reveal whether GC formation is affected in TASL KO mice which will strengthen the manuscript.

(5) The role of TASL in autoimmunity development was previously shown in chemically induced lupus model. Therefore, to increase the significance and novelty of this manuscript, the authors should investigate the B cell-intrinsic role of TASL using the BM chimera approach, especially the authors already performed BM chimera experiments in this manuscript.

Reviewer #2 (Betsy Barnes):

The short report by Johnstone et al. is a natural extension of the literature on TASL and IRF5, presenting expected results that indicate Taslko and Irf5ko mice phenocopy each other regarding B cell phenotypes and protection from lupus (PMCID: PMC11760370). As such, it is not clear to this Reviewer that the scientific findings, while strong and expected, fulfill the requirements of the journal - Short Reports should be novel, provocative and of general interest, in such a way as to spur future research. Additional comments are listed below.

1. Suppl. Fig1, a significant increase in T2 cells and pDC numbers from Taslko mice are shown (panels G and M). While relatively small, does this phenotype overlap with Irf5ko mice? Authors should discuss.

2. Fig. 1, why the discrepancy in CD69 and CD86 expression in response to CpG stimulation? This i interesting but should be discussed. Also to this point, why do the authors think most effects are seen downstream of CpG rather than IMQ?

3. Data from male mice in Fig. 2d and f is curious since there appears to be no change in plasmablast percents and numbers yet there is a significant decrease in IgG2c. Could the authors please discuss this discrepancy? Is there a secretion defect? The authors may need to examine intracellular levels and ELISPOT would be more rigorous to examine on a per cell basis.

4. Suppl Fig3a, it is very curious to see reduced levels of IgGs at steady state, which needs to be discussed. Is this similar to Irf5ko mice?

5. Fig 2H, what affect does NP-Ficoll have on the generation of PBs, given the conserved decrease in NP-IgM in both male and female Taslko mice?

6. Similar to comment 2 above, the authors conclude that Tasl is required for effective transduction of endolysosomal signaling yet only show compelling data for TLR9, with minimal effects in response to TLR7 or TLR4 signaling. This is confusing and is not supported by the data shown.

7. The authors need to discuss similarities and differences between their findings and findings in Irf5ko since the pathways overlap and similar experiments have been performed on Irf5ko mice (PMID: 39572974). This is particularly relevant since the authors end the discussion with a conclusion regarding TLR7-IRF5 signaling while the data support little effect of Tasl in response to the TLR7 ligand IMQ.

8. Some of the representative flow plots do not match data in bar graph. For example, Fig1c shows 92% IgD+ cells from Taslko while the bar graph in panel d shows 50-60%. The same is true for Fig2c and d; 3.1% plasmablasts are shown in Taslko mice while the bar graph shows all values to be below 2%.

Reviewer #3:

In the study by Johnstone et al., the data presented in the manuscript adds to the growing evidence that TASL and altered endosomal signalling is a key pathogenic mechanism in SLE by modulating B cell activation. The data are well presented and conducted, and the study will be a robust addition to the field as a short report. A few minor clarifications are needed prior to publication:

1 - Figure 1 is the only figure that does not have data from both sexes (female only), do the authors see sex differences on the impact of TASL KO on CpG/TLR stim?

2 - Please add male and female labels on the graphs themselves rather than just in the legend for Figure 3 for ease of interpretation.

3 - I suggest to combine the two section in the manuscript focused on mrl.lpr data into one as they are currently very short.

4 - Please add to the legend the sex of the representative images in Figure 4a.

5 - As the authors importantly highlight in the discussion, that due to the challenges they are using global TASL KO rather than conditional system and therefore in vivo effects could be driven (or at least) partially by changes in myeloid compartment (pDC). Do they authors see changes in this subset including IFN signalling in the lupus model? This may be out of scope for a short report, but it should at least be added as a limitation of study that they have not enumerated/phenotyped the myeloid compartment in these experiments.

---

## [Editor Report · Decision Letter 2]

4 Feb 2026

Dear Dr Clarke,

Thank you for your patience while we considered your revised manuscript "TASL is required for age-related B cell formation and autoimmunity" for publication as a Short Reports at PLOS Biology. This revised version of your manuscript has been evaluated by the PLOS Biology editors, and the Academic Editor.

Based on our Academic Editor's assessment of your revision, we are likely to accept this manuscript for publication, provided you satisfactorily address the remaining editorial points. Please also make sure to address the following data and other policy-related requests.

1) We routinely suggest changes to titles to ensure maximum accessibility for a broad, non-specialist readership, and to ensure they reflect the contents of the paper. In this case, we would suggest a minor edit to the title, as follows. Please ensure you change both the manuscript file and the online submission system, as they need to match for final acceptance:

"The adaptor protein TASL is required for age-related B cell emergence and lupus-like disease development in mice"

2) Unfortunately, Short Reports have a limit of 4 main figures; currently you have 5. Please rearrange them to reduce the number, either by combining them or sending one to the supplementary.

3) Please add weblink to funding agencies in the Financial Disclosure statement in the manuscript details during resubmission and within the manuscript.

Please supply the numerical values either in the a supplementary file or as a permanent DOI’d deposition for the following figures:

Figure 1ABDEG-N, 2BD-L, 3BCFG, 4BCFHJLM, 5BCEF, S1B-O, S2A-EG, S3A

5) Please cite the location of the data clearly in all relevant main and supplementary Figure legends, e.g. “The data underlying this Figure can be found in S1 Data” or “The data underlying this Figure can be found in https://doi.org/10.5281/zenodo.XXXXX”

6) We require the original, uncropped and minimally adjusted images supporting all blot and gel results reported in the Figure S1A

-- We will require these files before a manuscript can be accepted so please prepare and upload them now. Please carefully read our guidelines for how to prepare and upload this data: https://journals.plos.org/plosbiology/s/figures#loc-blot-and-gel-reporting-requirements

7) For figures containing FACS data (Figures 1CF, 2AC, 3AE, 4EGIK, 5A, S1DFJLNO), please provide the FCS files and a picture showing the successive plots and gates that were applied to the FCS files to generate the figure. We ask that you please deposit this data in the an Open Repository like Zenodo and provide the accession number/URL of the deposition in the Data Availability Statement in the online submission form.

8) Please add a scale bar in the following microscopy pictures in Figure 5D

9) Please ensure that your Data Statement in the submission system accurately describes where your data can be found and is in final format, as it will be published as written there

10) Per journal policy, if you have generated any custom code during the course of this investigation, please make it available without restrictions. Please ensure that the code is sufficiently well documented and reusable, and that your Data Statement in the Editorial Manager submission system accurately describes where your code can be found. More information on our Code Policy, what and how to share can be found here: https://journals.plos.org/plosbiology/s/code-availability

We expect to receive your revised manuscript within two weeks.

*Published Peer Review History*

*Press*

Sincerely,

Melissa

Melissa Vazquez Hernandez, Ph.D.

Associate Editor

PLOS Biology

---

## [Editor Report · Decision Letter 3]

18 Feb 2026

Dear Alex,

Thank you for the submission of your revised Short Reports "The adaptor protein TASL is required for age-related B cell emergence and lupus-like disease development in mice" for publication in PLOS Biology. On behalf of my colleagues and the Academic Editor, Takeshi Tsubata, I am pleased to say that we can in principle accept your manuscript for publication, provided you address any remaining formatting and reporting issues. These will be detailed in an email you should receive within 2-3 business days from our colleagues in the journal operations team; no action is required from you until then. Please note that we will not be able to formally accept your manuscript and schedule it for publication until you have completed any requested changes.

PRESS

Sincerely,

Melissa

Melissa Vazquez Hernandez, Ph.D., Ph.D.

Associate Editor

PLOS Biology
